# The Abundant Distribution and Duplication of SARS-CoV-2 in the Cerebrum and Lungs Promote a High Mortality Rate in Transgenic hACE2-C57 Mice

**DOI:** 10.3390/ijms25020997

**Published:** 2024-01-13

**Authors:** Heng Li, Xin Zhao, Shasha Peng, Yingyan Li, Jing Li, Huiwen Zheng, Yifan Zhang, Yurong Zhao, Yuan Tian, Jinling Yang, Yibin Wang, Xinglong Zhang, Longding Liu

**Affiliations:** Institute of Medical Biology, Chinese Academy of Medical Sciences and Peking Union Medical College, Kunming 650118, China; liheng@imbcams.com.cn (H.L.); xz479633385@163.com (X.Z.); shashap777@163.com (S.P.); yingyan0824@163.com (Y.L.); sola@imbcams.com.cn (J.L.); zhenghuiwen@imbcams.com.cn (H.Z.); zhangyifan10a@163.com (Y.Z.); yrrogerzhao@gmail.com (Y.Z.); m18859277876@163.com (Y.T.); yangjl0117@imbcams.com.cn (J.Y.); wangyb0027@163.com (Y.W.); zhangxinglong@imbcams.com.cn (X.Z.)

**Keywords:** SARS-CoV-2, cerebrum, high mortality rate, C57BL/6Smoc-*Ace2^em3(hACE2-flag-Wpre-pA)Smoc^* mice, Syrian hamsters, viral distribution and duplication

## Abstract

Patients with COVID-19 have been reported to experience neurological complications, although the main cause of death in these patients was determined to be lung damage. Notably, SARS-CoV-2-induced pathological injuries in brains with a viral presence were also found in all fatal animal cases. Thus, an appropriate animal model that mimics severe infections in the lungs and brain needs to be developed. In this paper, we compared SARS-CoV-2 infection dynamics and pathological injuries between C57BL/6Smoc-*Ace2^em3(hACE2-flag-Wpre-pA)Smoc^* transgenic hACE2-C57 mice and Syrian hamsters. Importantly, the greatest viral distribution in mice occurred in the cerebral cortex neuron area, where pathological injuries and cell death were observed. In contrast, in hamsters, viral replication and distribution occurred mainly in the lungs but not in the cerebrum, although obvious ACE2 expression was validated in the cerebrum. Consistent with the spread of the virus, significant increases in IL-1β and IFN-γ were observed in the lungs of both animals. However, in hACE2-C57 mice, the cerebrum showed noticeable increases in IL-1β but only mild increases in IFN-γ. Notably, our findings revealed that both the cerebrum and the lungs were prominent infection sites in hACE2 mice infected with SARS-CoV-2 with obvious pathological damage. Furthermore, hamsters exhibited severe interstitial pneumonia from 3 dpi to 5 dpi, followed by gradual recovery. Conversely, all the hACE2-C57 mice experienced severe pathological injuries in the cerebrum and lungs, leading to mortality before 5 dpi. According to these results, transgenic hACE2-C57 mice may be valuable for studying SARS-CoV-2 pathogenesis and clearance in the cerebrum. Additionally, a hamster model could serve as a crucial resource for exploring the mechanisms of recovery from infection at different dosage levels.

## 1. Introduction

Although public cautionary measures have been relaxed with the passage of time and improvements in treatment for COVID-19, at least 772,838,745 confirmed cases and 6,988,679 deaths have occurred worldwide as of 22 December 2023 (https://www.who.int/publications/m/item/covid-19-epidemiological-update---22-december-2023, accessed on 5 January 2024). Vaccine strategies can reduce the pathogenicity of SARS-CoV-2, however, until now, no vaccine has been able to induce sterilizing immunity. SARS-CoV-2 reinfection is most likely to occur after a short period of recovery from infection or vaccination, when the protection provided by the immune response is insufficient. Thus, SARS-CoV-2 remains a potentially lethal threat, especially to those who are immunocompromised, and countermeasures to prevent and treat COVID-19 are still a global health priority. Although the main clinical manifestations of COVID-19 are associated with respiratory or intestinal symptoms, reports of neurological signs and symptoms are increasing [1,2,3]. The primary neurologic symptoms include ‘brain fog’ (81%), headache (68%), numbness/tingling (60%), dysgeusia (59%), and anosmia (55%); most patients (85%) also report fatigue [4]. Importantly, histopathological changes in the central nervous system (CNS) were validated in fatal COVID-19 [5].

SARS-CoV-2 is a highly transmissible pathogen with broad tissue tropism, and it has been demonstrated that ACE2 polymorphisms can modulate susceptibility to SARS-CoV-2 [6,7,8]. Importantly, the emergence of SARS-CoV-2 variants, particularly the Omicron variant of concern (VOC), continue to pose a significant health threat due to their increased transmissibility and ability to evade the immune response [9,10,11]. Despite acquiring numerous mutations, all these Omicron subvariants still exploit ACE2 as the host receptor [12,13,14]. In human transcriptomes, ACE2 expression levels were highest in the small intestine, testis, kidneys, heart, thyroid, and adipose tissue, and interestingly, ACE2 expression could be detected in multiple brain regions at low levels [15,16,17]. In a few COVID-19 autopsy cases, the cerebellum (*n* = 3 of 24) was positive for SARS-CoV-2 [5]. It has been speculated that SARS-CoV-2 can enter the nervous system by crossing the neural–mucosal interface in the olfactory mucosa, exploiting the close vicinity of olfactory mucosal, endothelial, and nervous tissue [5]. Of course, the neurotropism of SARS-CoV-2 and its potential mechanisms of CNS entry and viral distribution need to be simulated in the appropriate animal models and validated in detail in terms of pathophysiology and immune response mechanisms.

The patterns of viral distribution and ACE2 expression in the brain in various animal models were sorted and compared, and we detected clear viral distribution in the brain in humanized ACE2 transgenic mice, including K18-hACE2 mice [18], HFH4-hACE2 mice [19], and knock-in mice [20]. To date, no data have been reported for other animals, although minor ACE2 expression at the mRNA or protein level was detected in all animals. Importantly, lethality was also found only in humanized ACE2 transgenic mice, including K18-hACE2 mice [18] and HFH4-hACE2 mice [19]; in fact, lethality was consistent with the viral distribution in the brain. However, the respiratory tract was still the main site of viral distribution in these mice, and infections in the brain were observed only in the deceased mice, suggesting that neuroinvasion is sporadic in these transgenic mice. As a result, there is an urgent need for an animal model with stable viral distribution and duplication in the brain.

In humanized ACE2 transgenic mice, hACE2 was validated to be expressed in most internal organs, including the respiratory tract and cerebrum, and these mice were susceptible to SARS-CoV-2 infection. Although only a minor level of hACE2 expression was detected in the brain in K18-hACE2 [18] and Hfh4-hACE2 mice [19], SARS-CoV-2 infection led to neuroinvasion and abundant viral distribution in the brain, which may be related to high lethality. K18-hACE2 mice replicate the virus to high titers in the nasal turbinates, lung and brain, and there was a survival advantage in the female mice, with 60% surviving infection, whereas all male mice succumbed to disease. Notably, brain infection was not observed in most of the animals at 3 dpi but was prevalent in K18-hACE2 mice necropsied at 5–11 dpi. Moreover, despite substantial N staining in the brains of K18-hACE2 mice, no obvious pathology was noted [18]. Similarly, HFH4-hACE2 mice expressed hACE2 at high levels in the lung but at varying expression levels in other tissues, including the brain, and obvious viral distribution was present in the brain in some mice. Importantly, in K18-hACE2 and HFH4-hACE2 mice, only deceased mice exhibited SARS-CoV-2 neuroinvasion of the brain [19]. On the other hand, in the knock-in mice, significant hACE2 expression was detected in the lung, small intestine, spleen, and kidney, and there was minor expression in the brain as evaluated via mRNA detection. Interestingly, robust viral RNA replication was observed in brain tissues; however, no pathological injuries or inflammatory responses were observed in the brain, and concordantly, no mortality was observed in this model [20]. In conclusion, all hACE2 mice intranasally challenged with SARS-CoV-2 developed interstitial pneumonia; a concern is that SARS-CoV-2 mainly replicates in the lungs of mice, although it may also target the brain, and infection in the brain may directly promote high mortality. In addition, the male mice showed much higher mortality than the female mice when the virus was abundantly distributed in the brain.

Notably, the SARS-CoV-2-binding domain of ACE2 presents a high degree of similarity between hamster and human receptors [21,22], suggesting the utility of golden Syrian hamsters in SARS-CoV-2 studies. Interestingly, most cerebral cortex neurons in hamsters exhibited positive staining for ACE2 [23]; this finding might facilitate further investigations of possible neural tissue damage in hamsters, although to date, there have been no reports on the pathophysiology of this disease in the brain. Moreover, the presence of high viral loads in the respiratory tract of hamsters, combined with an acute course of infection followed by rapid viral clearance and rapid recovery in the pathological injuries by 10–14 dpi [24,25], closely mirrors SARS-CoV-2 infection in humans. The pathological injuries in the lungs of hamsters are consistent with human infections, including interstitial to broncho-interstitial pneumonia, alveolar hemorrhage, and granulocyte infiltration. Overall, due to the high susceptibility to SARS-CoV-2, similarity to human pathology and ACE2 expression in the brain, hamsters could be a candidate model system in which to study the pathophysiology of SARS-CoV-2 infection in the brain.

On the basis of these reports, we used hamsters and C57BL/6Smoc-*Ace2^em3(hACE2-flag-Wpre-pA)Smoc^* hACE2 transgenic mice to research the viral distribution and duplication of SARS-CoV-2 in the cerebrum, to attempt to identify a stable animal model of infection in the cerebrum, and to study the development of pathophysiological and immune response mechanisms. Herein, we infected hamsters and hACE2-C57 mice at high and low dosages to detect the distribution and duplication of the virus in various tissues, especially in the cerebrum and respiratory tract. We found that the cerebrum and lungs were the major infection sites. We observed abundant distribution and duplication of SARS-CoV-2 at these sites in hACE2-C57 mice, while no obvious viral distribution or duplication was present in the cerebrum of the hamsters. Moreover, we found that SARS-CoV-2 in the cerebrum primarily appeared in the cerebral cortex neurons and particularly the pyramidal cell layer. In summary, both C57BL/6Smoc-*Ace2^em3(hACE2-flag-Wpre-pA)Smoc^* mice and hamsters were very susceptible to SARS-CoV-2 infection, and hACE2-C57 mice could be used as a stable animal model in which both the cerebrum and the lungs were major infection sites with abundant viral distribution and duplication, while hamsters could be a good animal model to research the recovery mechanisms because they had zero mortality at various infection dosages.

## 2. Results

### 2.1. Both hACE2-C57 Mice and Hamsters Were Susceptible to SARS-CoV-2 after Intranasal Infection, Which Resulted in Complete Fatality with High Viral Loads in the Cerebrums of hACE2-C57 Mice

We compared the infection dynamics by weight changes, survival proportions, viral shedding, and viral load in various tissues between hACE2 mice and hamsters. Both groups of animals were intranasally infected with SARS-CoV-2 at high and low dosages. hACE2 mice were infected with 1 × 10^2^ or 1 × 10^3^ CCID_50_, while hamsters were infected with 1 × 10^3^ or 1 × 10^5^ CCID_50_. We found that there was no obvious difference in clinical symptoms, survival rates, viral shedding, or viral load in various tissues between the animals infected with high and low dosages (Figure 1). The body weights of all the animals began to decrease at 3 dpi. In hamsters, body weight decreased to 92.14% of the baseline weight at 5 dpi and then began to recover, while the body weight of hACE2 mice decreased quickly beginning at 3 dpi (Figure 1A). All of the hACE2 mice, including mice with both high and low infection dosages, died before 6 dpi (Figure 1B), after their weight decreased to approximately 80% (Figure 1A). Notably, signs of neurological deficits (Appendix A), such as hind limb paralysis, head tilting, and tremors, were observed before death. Viral shedding was monitored daily via nasal washes and oropharyngeal swabs, and the dynamic trends were similar; the level of shedding peaked at 2 dpi and then decreased (Figure 1C). Notably, the shedding loads in hamsters were greater than those in hACE2 mice (Figure 1C), while the viral loads in most internal organs were similar between the two animal models, excluding the viral loads in the cerebrum of hACE2 mice, which were much greater than those in hamsters (Figure 1D), although the infection dosage in hamsters was 100-fold greater. The viral loads in all internal organs of both animals were measured, and robust viral RNA loads were detected in the cerebrum (in hACE2 mice), lungs, nasal mucosa, trachea, pulmonary lymph nodes, nasal lavage fluid, and bronchoalveolar lavage fluid, whereas little or no viral RNA was detected in the small intestine, heart, liver, spleen, and kidney (Figure 1D). The highest viral load in the internal organs, including the respiratory tract, was observed at 3 dpi, and the viral load in the cerebrums of hACE2 mice continued to increase until death (5 dpi), while the viral load in the cerebrum of hamsters was low during SARS-CoV-2 infection (Figure 1D). In summary, we concluded that both hACE2-C57 mice and hamsters were susceptible to SARS-CoV-2 upon intranasal infection, and the infection process was similar between high and low infection dosages, although there were slightly greater shedding rates and viral loads before 2 dpi in the groups infected with high dosages. Interestingly, a high viral load was present in the cerebrum of hACE2-C57 mice, along with 100% fatality, while lower viral loads were present in the cerebrum of hamsters (Figure 1D). On the basis of these data, we arranged an in-depth validation in which we selected the groups infected with high dosages to research viral distribution and duplication as well as pathological injuries.

### 2.2. ACE2 Was Expressed in Most Internal Organs of Both hACE2-C57 Mice and Hamsters, and the SARS-CoV-2 Viral N Protein Was Distributed in the Cerebrum of hACE2-C57 Mice but Not in That of Hamsters

The distribution and abundance of ACE2 and SARS-CoV-2 were analyzed via western blotting and immunofluorescence in all internal organs of hACE2-C57 mice and hamsters due to the important roles of ACE2 in SARS-CoV-2 infection (Figure 2). In hACE2-C57 mice, abundant ACE2 protein expression was detected via western blotting in the lung, nasal mucosa, kidney, and small intestine, and clear ACE2 protein signals were detected in the spleen, trachea, liver, and cerebrum but not in the heart (Figure 2A). In hamsters, abundant ACE2 protein was validated via western blotting in the lung, kidney, nasal mucosa, and small intestine, and clear ACE2 protein signals were observed in the liver and cerebrum, but no ACE2 protein was detected in the heart, spleen, or trachea (Figure 2A). In addition, the ACE2 distribution was validated via immunofluorescence, and a minor amount of ACE2 was observed in the spleens of the hamsters (Figure 2B–D and Appendix A). Then, the colocalization of ACE2 and the SARS-CoV-2 N protein was detected via immunofluorescence (Figure 2B–D). Interestingly, we found that the abundances of the ACE2 and SARS-CoV-2 N proteins were inconsistent in some internal organs. ACE2 was distributed widely in the cerebrum at low levels, and many SARS-CoV-2 N proteins were present at 3 dpi and 5 dpi in hACE2-C57 mice, while no obvious N protein was present in the cerebrum of hamsters (Figure 2B). In the respiratory tract, both the ACE2 and N proteins were abundant in the lung and nasal mucosa during SARS-CoV-2 infection (Figure 2C,D). In addition, the ACE2 protein was abundant in the submucosa of hACE2-C57 mice but scarce in that of hamsters, while the SARS-CoV-2 N protein was mostly distributed in the mucous membranes of both animals, and ACE2 expression was limited (Figure 2D). Moreover, abundant ACE2 protein was present in the kidney, small intestine, and spleen (Figure 2A and Appendix A); however, few SARS-CoV-2 viruses were found in these locations (Figure 1D). According to these results, we found that SARS-CoV-2 was abundant in the respiratory tracts and cerebrum of hACE2-C57 mice, while ACE2 levels were lower in the cerebrum than in the lungs and nasal mucosa. SARS-CoV-2 was abundant in the respiratory tract but scarce in the cerebrum of hamsters, although obvious ACE2 was present in the cerebrums. Therefore, we concluded that the presence of ACE2 was primary for SARS-CoV-2 infection; however, the abundances of ACE2 and SARS-CoV-2 were not consistent in the infection process.

### 2.3. SARS-CoV-2 Was Mainly Distributed in the Neurons of the Cerebral Cortex of hACE2-C57 Mice, While the Virus Was Widely Detected in Alveolar Cells in the Lungs of Both hACE2-C57 Mice and Hamsters

The abundance and distribution of SARS-CoV-2 were validated via immunohistochemistry (IHC) in the cerebrum and lungs of hACE2-C57 mice and hamsters. In the cerebrum, abundant SARS-CoV-2 N protein was present in hACE2-C57 mice but not in hamsters, and importantly, we demonstrated that the SARS-CoV-2 N protein was abundantly distributed in the pyramidal cell layer of the cerebral cortex (Figure 3A). In the lungs, abundant SARS-CoV-2 N protein was present in hACE2-C57 mice and hamsters, and interestingly, N protein was present around bronchioles at the beginning of infection (1 dpi). Thereafter, the N protein was distributed mainly in lung interstitial cells (Figure 3B). Overall, there was a high viral load in the cerebrum and lungs of hACE2-C57 mice from 3 dpi until death, and the viral load in the lungs of hamsters decreased from its peak to a low level at 10 dpi.

### 2.4. Significant Viral Replication and a Strong Viral Load Were Observed in Both the Cerebrums and Lungs of hACE2-C57 Mice, While Abundant Viral Replication Was Present in the Lungs but Not the Cerebrum of Hamsters

Viral duplication and distribution at the mRNA and protein levels were validated in the cerebrum and lungs throughout the infection process in hACE2-C57 mice and hamsters, and RNAscope–immunofluorescence codetection was employed. Probe-S-sense can recognize the antisense sequence of S-mRNA and reflects SARS-CoV-2 duplication because the antisense sequence is present only during viral duplication. Probe-S can recognize S-mRNA and reflects SARS-CoV-2 distribution at the mRNA level. In the cerebrum, obvious viral duplication (detected via Probe-S-sense) and abundant viral distribution (detected via Probe-S and Antibody-N) were present at 3 dpi and 5 dpi in hACE2-C57 mice, while no obvious viral duplication or distribution was present in the cerebrum of hamsters (Figure 4A). In the lungs, obvious viral duplication and abundant viral distribution were observed at 1 dpi, 3 dpi and 5 dpi in both hACE2-C57 mice and hamsters (Figure 4B). Interestingly, the viral duplication levels were consistent with the viral distribution in the cerebrum and lungs in both animals, and we concluded that the high viral loads distributed in the cerebrum of hACE2-C57 mice were not transported from other internal organs and might instead come from the high levels of replication and multiplication of SARS-CoV-2 present in the cerebrum.

### 2.5. Obvious Increases in IL-1β and Minimal Changes in IFN-γ Were Observed in the Cerebrum of hACE2-C57 Mice Infected with SARS-CoV-2

Proinflammatory molecules, including IL-1β and IFN-γ, were detected via immunofluorescence and real-time PCR in the cerebrum and lungs of hACE2-C57 mice and hamsters infected with SARS-CoV-2. We found that abundant IL-1β and a small amount of IFN-γ were present at 3 dpi in the cerebrum of hACE2-C57 mice, with more IL-1β and no IFN-γ present at 5 dpi; in contrast, there was no obvious presence of IL-1β or IFN-γ in the cerebrum of hamsters (Figure 5A,C). In the lungs of both animals, IL-1β and IFN-γ were conspicuously present at 3 dpi, and more IL-1β and little IFN-γ were present at 5 dpi (Figure 5B,C). To better understand the inflammatory response to SARS-CoV-2 infection, we investigated the mRNA expression levels of IFN-γ and IL-1β in the cerebrum and lungs during SARS-CoV-2 infection, and IL-1β and IFN-γ were expressed in the lungs of both mice and hamsters. The mRNA level of IL-1β was also high at 3 dpi or 5 dpi in the cerebrum of hACE2 mice but was low in that of hamsters (Figure 5D). According to these results, obvious IL-1β was present at 3 dpi early during infection and subsequently increased, and IL-1β reached its highest level in the cerebrum of hACE2-C57 mice until death; moreover, no obvious IL-1β was detected during end-stage infection in the lungs of hamsters. IL-1β may play important roles in the inflammatory response in the cerebrum and lungs in SARS-CoV-2 infection.

### 2.6. Prominent Pathological Injuries Were Observed in the Cerebrum of hACE2-C57 Mice, While Both hACE2-C57 Mice and Hamsters Exhibited Severe Interstitial Pneumonia following SARS-CoV-2 Infection

Pathological injuries in the cerebrum and lungs were detected and analyzed via hematoxylin and eosin (HE) staining in hACE2-C57 mice and hamsters during SARS-CoV-2 infection. In the cerebrum, conspicuous pathological injuries were present in the cerebrum of hACE2-C57 mice, especially at 5 dpi (the time of death). Specifically, at 3 dpi, there was some inflammatory cell infiltration and cell necrosis in the early stage, and many dead cells and vacuolar cavities were present in the cerebral cortex at 5 dpi when the mice died (Figure 6A). In the lungs, severe interstitial pneumonia presented in both hACE2-C57 mice and hamsters with SARS-CoV-2 infection, and the development process of interstitial pneumonia was similar in the two animal models. Obvious inflammatory cell infiltration, stromal cell hyperplasia and some necrocytosis were present at 3 dpi in early infection, and then, a large area of alveolar wall cell necrocytosis and inflammatory cell infiltration was present at 5 dpi, and there was a large amount of mucus and cells in the bronchioles (Figure 6B). Finally, hACE2-C57 mice died with conspicuous pathological injuries in the cerebrum and lungs, and the pathological injuries in the lungs recovered gradually at 10 dpi in hamsters.

## 3. Discussion

COVID-19 was initially thought to be a respiratory disease [26,27]; however, neurological complications have been described in human COVID-19 patients [1,2,3], although lung damage is the primary cause of death in most patients. Importantly, in a few COVID-19 autopsy cases, SARS-CoV-2 mRNA and protein and pathological injuries were found in the brain [5]. Moreover, obvious viral distribution in the brain was found in all animals that died; indeed, it was speculated that the viral distribution in the brain and the accompanying pathological injuries promoted lethality [18,19]. Thus, a compatible animal model should be developed to mimic infection in the cerebrum under experimental conditions. In this paper, we compared the SARS-CoV-2 infection dynamics and pathological injuries between C57BL/6Smoc-*Ace2^em3(hACE2-flag-Wpre-pA)Smoc^* transgenic hACE2-C57 mice and Syrian hamsters infected with high and low dosages of SARS-CoV-2, and we demonstrated that both the cerebrum and the lungs were major infection sites in hACE2-C57 mice. SARS-CoV-2 was abundantly distributed and duplicated presented in the cerebrum and lungs of hACE2-C57 mice in the high- and low-dose infection groups, while no obvious viral distribution or duplication was present in the cerebrum of hamsters. Moreover, we found that SARS-CoV-2 in the cerebrum primarily appeared in the pyramidal cell layer of the cerebral cortex. Concordantly, an obvious proinflammatory response and severe pathological injuries were present in the cerebrum of hACE2-C57 mice. High levels of viral distribution, IL-1β-associated inflammation and conspicuous pathological injuries in the cerebrum may promote susceptibility to SARS-CoV-2 infection and death in hACE2-C57 mice. In hamsters, abundant viral distribution and duplication also appeared in the lungs but not in the cerebrum; severe interstitial pneumonia presented from 3 dpi to 5 dpi and then gradually subsided, and the clinical symptoms recovered at 10–14 dpi. Consistent with previous reports, the whole infection process in hamsters was similar to the process that has been observed in humans. In summary, both C57BL/6Smoc-*Ace2^em3(hACE2-flag-Wpre-pA)Smoc^* and hamsters were very susceptible to SARS-CoV-2 infection, and the cerebrum and lungs were the major infection sites in hACE2-C57 mice.

Regarding the colocalization of ACE2 and SARS-CoV-2 N protein, we found no obvious viral distribution in the tissues in which there was no ACE2 expression, and the abundance of viral distribution was not consistent with ACE2 levels. For example, we observed abundant viral distribution and duplication in the cerebrum of hACE2-C57 mice, although the ACE2 levels in this site were low. ACE2 polymorphisms may alter human susceptibility to SARS-CoV-2 infection and contribute to ethnic and geographical differences in the spread of SARS-CoV-2 [8]. According to our results, SARS-CoV-2 was clearly distributed and duplicated in the cerebrum of hACE2-C57 mice, and we speculated that viral duplication and assembly occurred with high efficiency, while viral clearance proceeded with low efficiency in the cerebral cortex of hACE2-C57 mice. On the other hand, both hACE2-C57 mice and hamsters were susceptible to SARS-CoV-2 infection, and abundant viral distribution and duplication appeared in the nasal mucosa and lungs. Unexpectedly, abundant ACE2 protein was present in the kidney and spleen; however, no obvious viral distribution appeared in these tissues during SARS-CoV-2 infection in most animals. In particular, ACE2 protein was distributed in the cerebrum of both animals; however, abundant viral distribution was present in hACE2-C57 mice but not in hamsters. Therefore, the switch pathways and pathway of entry into the cerebrum or other tissues could be researched in detail in C57BL/6Smoc-*Ace2^em3(hACE2-flag-Wpre-pA)Smoc^* and hamsters.

More precise examination of the life cycle and pathophysiology of SARS-CoV-2 via the diversification of assays will improve the comprehension of virus features and pathogenetic mechanisms, and in fact, none of the models tested thus far completely reflect human COVID-19 [28]. We compared the viral shedding level, viral distribution and duplication, inflammatory response, and pathological injuries in various internal organs in hACE2-C57 mice and hamsters, and we found that the shedding levels in hamsters were higher than those in hACE2 mice, perhaps due to the different infection dosages; however, the viral loads in most internal organs were equivalent in both animal models, except that the viral loads in the cerebrum were much higher in hACE2 mice. We speculated that the two animal models had equal sensitivity to SARS-CoV-2 infection in the internal organs. Interestingly, a high level of viral distribution and duplication was present in the cerebrum of hACE2-C57 mice, with 100% fatality, while a low viral load was present in the cerebrum of hamsters; concordantly, severe pathological injuries and obvious inflammatory responses were present in the lungs of both animal models and in the cerebrum of hACE2 mice. Finally, all hACE2 mice died before 6 dpi with abundant viral distribution and pathological injuries in the lungs and cerebrum, while all hamsters recovered after 10–14 dpi and regained normal lung function. According to these results, we concluded that hACE2 mice can be used to evaluate the protective and immune effects of biological drugs or vaccines in the exacting terms, whereas hamsters could continue to be used for similar studies over longer periods; additionally, hamsters could be beneficial for research on recovery mechanisms throughout the infection process. In fact, we found that several prevaccine and human neutralizing antibodies could protect hACE2 transgenic mice from lethal SARS-CoV-2 challenge with robust viral clearance [29,30].

In previous reports, hamsters were infected with 8 × 10^4^ CCID_50_ [24], 1 × 10^5^ CCID_50_ [25] or 1 × 10^6^ PFU [31]; ferrets were infected with 1 × 10^5.5^ CCID_50_ [32]; K18-hACE2 mice [18] were infected with 1 × 10^4^, 2 × 10^4^ or 2 × 10^3^ PFU; HFH4-hACE2 mice were intranasally infected with 3.3 × 10^4^ CCID_50_ [19]; knock-in mice were infected with 4 × 10^5^ CCID_50_ [20]; C57BL/6Smoc-*Ace2^em3(hACE2-flag-Wpre-pA)Smoc^* hACE2 mice were infected with 1 × 10^4^ CCID_50_ [30] or 1 × 10^3^ PFU [29]; and rhesus monkeys were infected with 1 × 10^6^ CCID_50_ [33]. According to these data, we infected the hACE2 mice intranasally by high and low infection dosages of 1 × 10^2^ CCID_50_ and 1 × 10^3^ CCID_50_, while hamsters were infected by high and low infection dosages of 1 × 10^3^ CCID_50_ and 1 × 10^5^ CCID_50_, and susceptibility to SARS-CoV-2 infection was evaluated in both animal models. Interestingly, there was no obvious difference in the clinical symptoms, survival proportions, viral shedding, or viral load in various tissues between the groups infected with high and low dosages. In fact, another group of hamsters was infected with 1 × 10^6^ CCID_50_, and all the viral shedding, viral loads in various tissues, and pathological injuries were similar to those of the groups infected with 1 × 10^3^ CCID_50_ and 1 × 10^5^ CCID_50_; therefore, the data are not shown here due to their similarity. We concluded that both hACE2 mice and hamsters were susceptible to SARS-CoV-2 infection.

Neuroinvasion by SARS-CoV-2 has been demonstrated in K18-hACE2 mice [18,34], HFH4-hACE2 mice [19], knock-in mice [20], and newly weaned hamsters in multiple previous reports, and SARS-CoV-2 can enter the nervous system by crossing the neural–mucosal interface in the olfactory mucosa according to studies of autopsy material. Interestingly, pathological damage in the brain was validated in K18-hACE2 mice via TEM [34]. Consistently, lethality was also found only in humanized ACE2 transgenic mice, including K18-hACE2 mice and HFH4-hACE2 mice, and we concluded that lethality was consistent with the viral distribution in the brain. However, the respiratory tract was still the main site of viral distribution in these mice, and infections in the brain were observed only in the deceased mice, suggesting that neuroinvasion is not widespread in these transgenic mice. Interestingly, we generated a different animal model, C57BL/6Smoc-*Ace2^em3(hACE2-flag-Wpre-pA)Smoc^* mice here, in which abundant distribution and duplication of SARS-CoV-2 were detected in both the cerebrum and lungs via direct evidence from RNAScope and IHC; moreover, proinflammatory cytokines and pathological damage in the cerebrum were detected in situ. Consistent with previous data, infection in the cerebrum may be the major factor promoting lethality, given that all the transgenic hACE2-C57 mice died at high or low infection doses.

Importantly, we found that high levels of IL-1β were present in the cerebrum after SARS-CoV-2 infection in transgenic mice, while the levels were low in hamsters. IL-1β is one of the most powerful proinflammatory cytokines against infection and regulates the expression of several molecules involved in inflammation [35,36]. In patients infected with SARS-CoV-2, increased levels of proinflammatory cytokines, including IL-1β, were found, and these increased cytokines and chemokines mediated infection immunopathogenesis and played important roles in the progression of COVID-19 [37,38]. An increase in IL-1β expression during SARS-CoV-2 infection was demonstrated in the serum of hACE2-aged mice [20] and in the nasal mucosa of hamsters [25]. Interestingly, increased IL-1β expression during SARS-CoV-2 infection was found in both the lungs and brain of newly weaned hamsters, and neuroinvasion of SARS-CoV-2 was demonstrated [39]. We found clear expression of IL-1β early during infection at 3 dpi, but these levels increased later; IL-1β reached its highest level in the cerebrum of hACE2-C57 mice before death, and no obvious increase in IL-1β was detected in the hamsters. In addition, significant increases in IL-1β and IFN-γ were observed in the lungs of both animals. We concluded that IL-1β may play important roles in the inflammatory response in the cerebrum and lungs during SARS-CoV-2 infection and that at high levels of viral distribution, IL-1β-associated inflammation and conspicuous pathological injuries in the cerebrum may promote susceptibility to SARS-CoV-2 infection and death in hACE2-C57 mice. In summary, we used hamsters and C57BL/6Smoc-*Ace2^em3(hACE2-flag-Wpre-pA)Smoc^* transgenic hACE2 mice to research viral distribution and duplication in SARS-CoV-2 infection, and we found for the first time that the cerebrum and lungs were the major infection sites in hACE2 mice, with abundant viral distribution and duplication. Mice and hamsters are ideal models for the study of COVID-19 due to their small size, ready availability, low cost of care, availability as specific-pathogen-free (SPF) animals, and in-depth characterization across a variety of translational models, as well as the commercial availability of many antibodies and tools for immunological studies in these animals [40,41,42]. We also found that the inflammatory brain damage in the newly weaned hamsters was very interesting, whereas in our study, the major locations of SARS-CoV-2 infection and pathological damage were the respiratory tracts in adult hamsters. Hamsters and mice were tested in parallel, which strengthened the results and conclusions about the correlation between lethality and infection in the cerebrum. According to our results, hACE2-C57 mice could be used as an animal model of severe COVID-19, in which lethality is high and both the cerebrum and lungs are major sites of viral distribution, to research pathophysiological mechanisms and evaluate drug development; meanwhile, hamsters could serve as a good animal model to research the mechanisms of recovery from various infection dosages due to their mortality rate of zero and their strong recovery abilities.

## 4. Materials and Methods

### 4.1. Animals and Biosafety

Female hACE2 mice aged from 6 to 8 weeks were purchased from Shanghai Model Organisms Center, Inc. (Shanghai, China); the specific name of the line was C57BL/6Smoc-*Ace2^em3(hACE2-flag-Wpre-pA)Smoc^*. Female Syrian hamsters aged from 6 to 8 weeks were purchased from Beijing Vital River Laboratory Animal Technology Co., Ltd. (Beijing, China). In this paper, Syrian hamsters are simply referred to as “hamsters” for brevity. All animal experiments were conducted under prior approval from the Animal Ethics Committee of the Institute of Medical Biology, IMBCAMS, according to the National Guidelines on Animal Work in China, and the relevant project identification codes were DWSP202207011 and DWSP202107015. To minimize pain, all experiments were carried out in strict accordance with national guidelines for animal welfare. In accordance with the principles of animal ethics, humane euthanasia was performed.

Two groups of C57BL/6Smoc-*Ace2^em3(hACE2-flag-Wpre-pA)Smoc^* mice and two groups of hamsters were used to research the viral dynamics and pathophysiology of SARS-CoV-2 infection at high and low infection dosages. After anesthesia, the animals were infected by nasal drip; the drip for hACE2 mice was 20 µL, and that for hamsters was 80 µL. The nasal cavities were washed 3 times with an equal volume of PBS after 12 h of dripping. The groups were as follows: hACE2 mice were infected with 1 × 10^2^ CCID_50_, hACE2 mice infected with 1 × 10^3^ CCID_50_, hamsters infected with 1 × 10^3^ CCID_50,_ and hamsters infected with 1 × 10^5^ CCID_50_. Twenty two animals were included in each group. First, 10 animals were infected and monitored for daily body weight changes, survival rates, and viral shedding in nasal lavage fluid and oropharyngeal swabs; oropharyngeal shedding was assessed via swabs, while nasal shedding was assessed via 20 µL nasal washes through a micropipette. The other 12 animals were infected for detection of viral distribution, viral duplication, inflammatory response, and pathological injuries at 1, 3, 5, and 10 dpi. All work with infectious SARS-CoV-2 was performed with approval under Biosafety Level 3 (BSL3) and Animal Biosafety Level 3 (ABSL3) conditions by the Institutional Biosafety Committee of the Institute of Medical Biology (IMB) in Kunming National High-level Biosafety Primate Research Center.

### 4.2. Viruses

The viral strain SARS-CoV-2-KMS1/2020 (GenBank accession number: MT226610.1) was isolated from sputum collected from a COVID-19 patient by the Chinese Academy of Medical Sciences (IMBCAMS) and propagated and titered on Vero cells in DMEM (Sigma-Aldrich, St. Louis, Darmstadt, Germany). The stock viruses were frozen at −80 °C and prepared for the following experiments.

### 4.3. Virus Load Detection

RNA from oropharyngeal swabs, 20 µL nasal washes, and 100 mg homogenized tissue was extracted using TRIzol reagent (Tiangen, Inc., Beijing, China) in 20 µL of RNA-free water, and 2 µL of total RNA was detected via RT-real-time PCR (One Step PrimeScript™ RT–PCR Kit (perfect real time), RR064A, Takara, Inc.,Dalian, China). The primers and probe used were as follows: E_Sarbeco_F: 5′-ACAGGTACGTTAATAGTTAATAGCGT-3′; E_Sarbeco_R: 5′-ATATTGCAGCAGTACGCACACA-3′; and E_Sarbeco-P: 5′-ACACTAGCCATCCTTACTGCGCTTCG-3′. For the quantification of viral RNA, a standard curve was generated using 10-fold dilutions of the RNA standard, and the standard curve was y =−0.2795x + 10.882.

### 4.4. Western Blotting

hACE2 mice and hamsters without infection were sacrificed and dissected, and all of the internal organs were homogenized in RIPA buffer. The samples were detected via western blotting to measure ACE2 expression. The primary antibody used was Rb mAb against human ACE2 (ab108209; Abcam, Cambridge, UK), and the primary antibody was incubated at a 1:1000 dilution overnight at 4 °C. Goat anti-rabbit IgG (HRP) (ab6721) was used for visualization.

### 4.5. Immunofluorescence (IF)

Paraffin-embedded tissue sections were dewaxed, antigen retrieval was performed, the tissue slides were permeabilized with 0.1% Triton X-100 for 15 min, and the tissue sections were subsequently blocked for 1 h in 5% BSA at room temperature (RT).

#### 4.5.1. Determination of the Distribution and Abundance of ACE2 and SARS-CoV-2 via Immunofluorescence

The sections were labeled with Rb mAb against human ACE2 (Abcam, ab108209) and the SARS-CoV-2 nucleocapsid antibody chimeric Mab (Sino Biological, Inc., Beijing, China. Cat: 40143-MM05TA) at a 1:500 dilution overnight at 4 °C. Finally, human ACE2 protein antigens were visualized via donkey anti-rabbit IgG H&L (Alexa Fluor^®^ 594) (Abcam, ab150076), and SARS-CoV-2 N protein antigens were visualized via goat anti-human IgG Fc (DyLight^®^ 488) (Abcam, ab97003) at a 1:500 dilution for one hour. The images were captured via a Leica TCS SP8 laser confocal microscope.

#### 4.5.2. Determination of the Distribution and Abundance of IFN-γ and IL-1β via Immunofluorescence

The sections were labeled with an IFN-γ polyclonal antibody (ImmunoWay, Suzhou, China. cat# YT2279) at a 1:500 dilution for 1 h at room temperature and visualized with an Opal 520 Fluorophore (Akoya Biosciences, Massachusetts, U.S.A. Opal 3-Plex Manual Detection Kit, REF: NEL810001KT). Antigen retrieval was subsequently performed. The tissue sections were blocked for 1 h in 5% BSA at room temperature and then labeled with the Rb pAb to IL-1 (Abcam, ab2105) at a 1:500 dilution overnight at 4 °C. Finally, the IL-1 protein antigens were visualized via incubation with donkey anti-rabbit IgG H&L (Alexa Fluor^®^ 594) (Abcam, ab150076) at a 1:500 dilution for one hour. The images were captured via a Leica TCS SP8 laser confocal microscope.

### 4.6. Immunohistochemistry (IHC)

The paraffin-embedded tissue sections were deparaffinized in xylene, rehydrated in a graded series of ethanol, and rinsed with double-distilled water. The sections were incubated with rabbit antiSARS-CoV-2 N antigen (Sino Biological, Beijing, China) for 1 h after heat-induced epitope retrieval. Antibody labeling was visualized via the development of DAB. Digital images were captured and evaluated via a histological section scanner (Pannoramic MIDI, 3D HISTECH, Budapest, Hungary).

### 4.7. RNAscope-I Codetection

RNAscope-IF codetection was performed using the RNAscope^®^ Multiplex Fluorescent v2 Assay combined with immunofluorescence-integrated codetection (ACD). Paraffin-embedded tissue sections were labeled with an anti-SARS-CoV-2 N protein antibody (Sino Biological, China) at a 1:500 dilution overnight at 4 °C. Then, ISH probes, including V-nCoV2019-S (ACD) and V-nCoV2019-S-sense (ACD), were hybridized to RNA, followed by amplification of the signal operation, and the RNAscope^®^ Multiplex Fluorescent v2 Assay was run to visualize SARS-CoV-2 N protein antigens via donkey anti-rabbit IgG H&L (Alexa Fluor^®^ 647) (Abcam, ab150075) at a 1:500 dilution. The images were captured via a Leica TCS SP8 laser confocal microscope.

### 4.8. Histopathology

hACE2 mice and hamsters infected with SARS-CoV-2 were sacrificed, dissected under ABSL3 conditions, and stored for 2 weeks with formalin fixation. Then, the samples were fixed for 2 h in 10% formalin, 1 h in 70% ethanol, 1 h in 80% ethanol, 1 h in 90% ethanol, 1 h in 95% ethanol 3 times, 1 h in xylene, 30 min in xylene, 30 min in paraffin, and 1 h in paraffin twice. After tissue embedding and slicing, the sections of paraffin-embedded tissue were deparaffinized in xylene, rehydrated in a graded ethanol series, and rinsed with double-distilled water. Then, the sections were exposed to hematoxylin for 15 min, water for 1 min, 1% HCl in ethanol for 5 s, water for 1 min, ammonium hydroxide for 10 s, water for 1 min, 0.5% eosin for 30 s, 75% ethanol for 10 s, 95% ethanol twice for 10 s each, ethanol twice for 10 s each, and xylene was added twice for 10 s each.

### 4.9. IFN-γ and IL-1β Quantifications via RT-Real Time PCR

The same point of cerebrum and lung were harvested for total RNA extraction, and the levels of IFN-γ and IL-1β were detected in the total RNA via real time PCR (One Step TB Green^®^ PrimeScript™PLUS RT-PCR Kit; Code No.RR096A; TAKARA), and the primer sequences were as follows: mice-IFN-γ (F: 5′-GCCACGGCACAGTCATTGA; R: 5′-TGCTGATGGCCTGATTGTCTT); mice-IL-1β (F: TGCCACCTTTTGACAGTGATG; R: 5′-AAGGTCCACGGGAAAGACAC); mice-Actin (F: GTGACGTTGACATCCGTAAAGA; R: GCCGGACTCATCGTACTCC); hamster-IFN-γ (F: 5′-TGCATCTTGGCTTTGTTGCTC; R: TCCCCTCCATTCACGACATC); hamster-IL-1β (F: GTGGACAACAAAGCTCGTGG; R: AGCCCGTCAACCTCAAAGAA); hamster-Actin (F: ATGGCCAGGTCATCACCATTG; R: CAGGAAGGAAGGCTGGAAAAG).

### 4.10. Statistical Analysis

The data were analyzed using GraphPad Prism 8.0.1 (244), and the data were also analyzed via one-way ANOVA using SPSS PASW statistical software version 18.0. * 0.01 < *p* ≤ 0.05, ** 0.001 < *p* ≤ 0.01, and *** *p* ≤ 0.001.

## Figures and Tables

**Figure 1 ijms-25-00997-f001:**
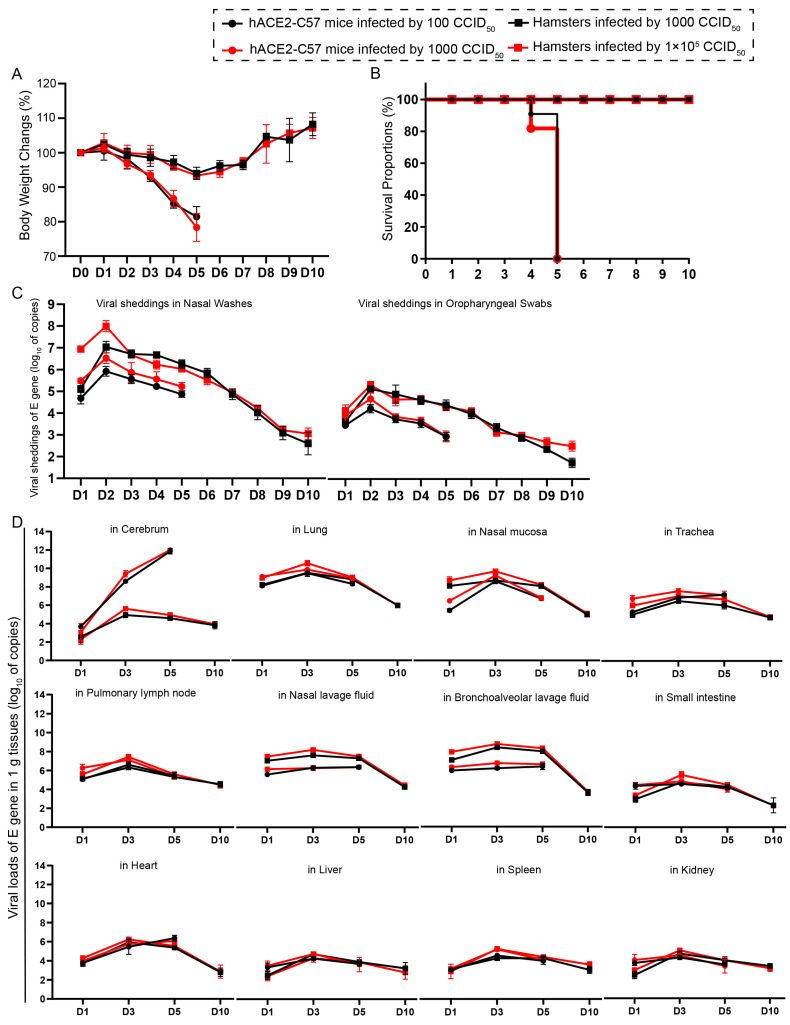
Both hACE2-C57 mice and hamsters exhibited susceptibility to SARS-CoV-2 following intranasal infection. (**A**) Body weight changes were monitored daily in hACE2-C57 mice and hamsters infected with SARS-CoV-2 at high and low dosages (*n* = 4). (**B**) The survival proportions were recorded in both animal models during infection. (**C**) Viral shedding in nasal washes and oropharyngeal swabs was assessed daily via detecting the E-gene of SARS-CoV-2 (*n* = 4). (**D**) The viral loads in all internal organs of both animal models were assessed throughout the infection process via detecting the E gene of SARS-CoV-2 (*n* = 3).

**Figure 2 ijms-25-00997-f002:**
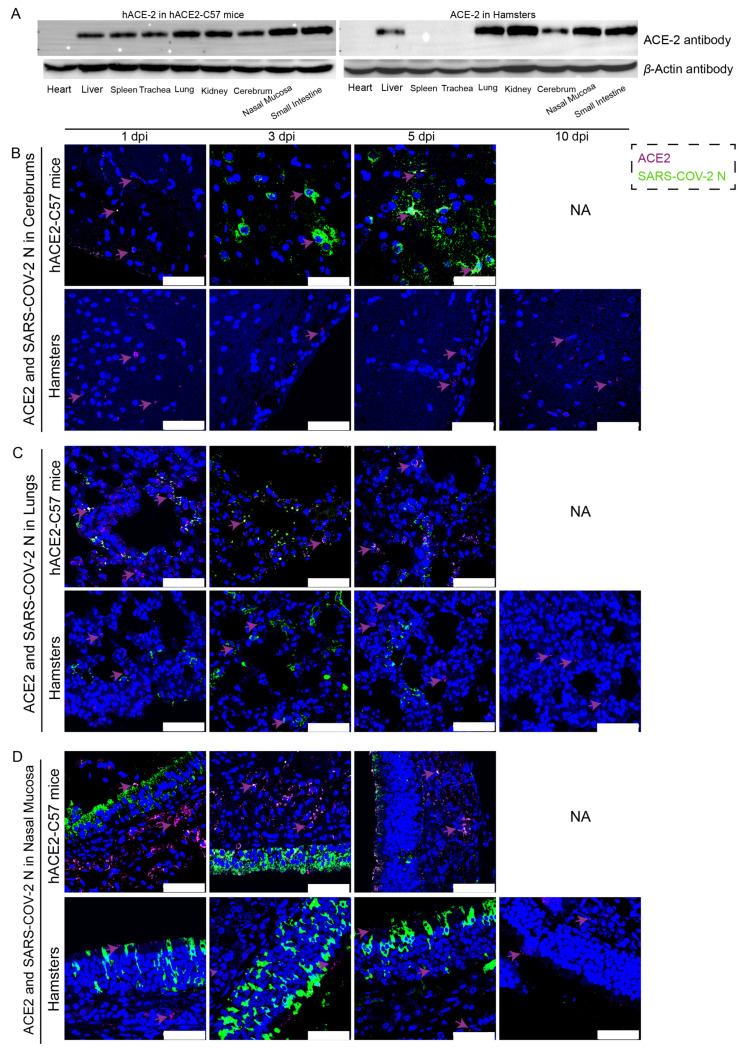
The primary sites of viral distribution were the cerebrum in hACE2-C57 mice and the respiratory tract in both animal models. (**A**) The distribution and abundance of ACE2 in most internal organs were validated via western blotting in hACE2-C57 mice and hamsters. (**B**–**D**) The colocalization of ACE2 and the SARS-CoV-2 N protein was detected in the cerebrum (**B**), lungs (**C**) and nasal mucosa (**D**) throughout the infection process; ACE2 is shown in purple and with purple arrows indicating representative signals, and the N protein is shown in green. NA indicates no data because all the mice died before 10 dpi. The scales in the pictures were 50 µm.

**Figure 3 ijms-25-00997-f003:**
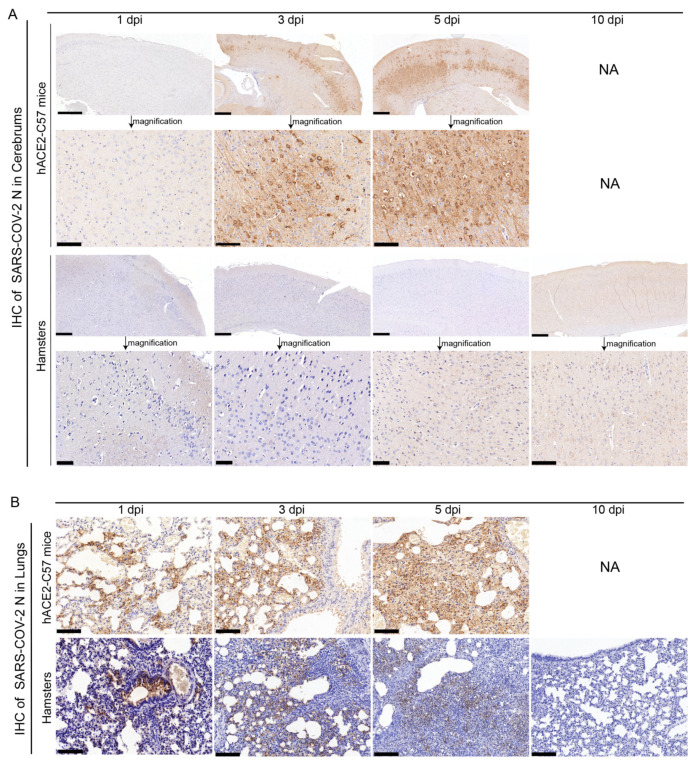
SARS-CoV-2 was detected mainly in the cerebral cortex neurons of hACE2-C57 mice. (**A**,**B**) The SARS-CoV-2 distribution detected via IHC staining of the N protein in the cerebrum (**A**) and lungs (**B**) of hACE2-C57 mice and hamsters throughout the infection process. NA indicates no data because all the mice died before 10 dpi. N proteins are visualized as brown via DAB staining during IHC staining, and the cell nuclei are visualized as blue via hematoxylin staining. The arrows in (**A**) present the pictures in below (scales were 100 µm) were enlarged from the pictures in top (scales were 400 µm). The scales in (**B**) were 100 µm.

**Figure 4 ijms-25-00997-f004:**
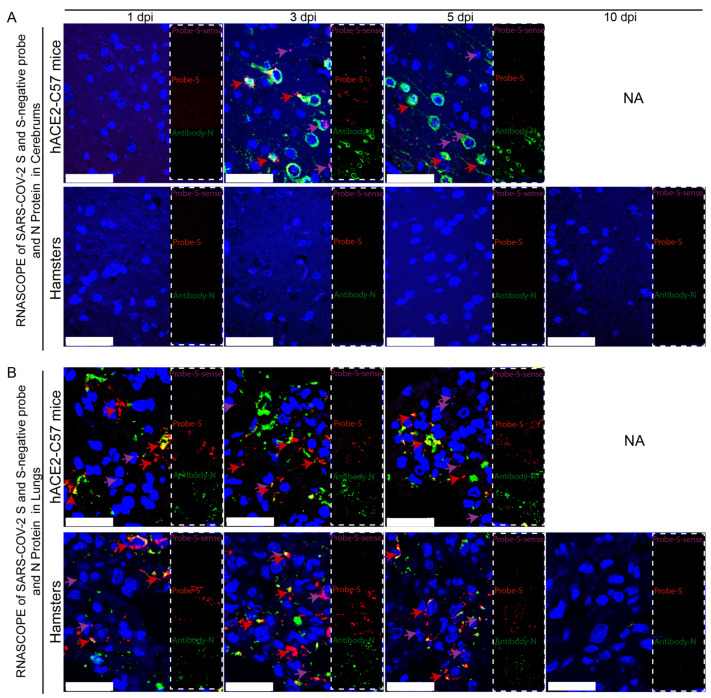
Obvious viral replication and robust viral distribution were present in the cerebrum and lungs of hACE2-C57 mice, while viral replication and distribution were present in the lungs but not in the cerebrum of hamsters. (**A**,**B**) Viral duplication and distribution were detected via RNAscope and immunofluorescence in the cerebrum (**A**) and lungs (**B**). Probe-S-sense reflects viral duplication at the RNA level and is shown in purple with purple arrows indicating representative signals; Probe-S reflects viral distribution at the RNA level and is shown in red with red arrows indicating representative signals; and Antibody-N reflects viral distribution at the protein level and is shown in green. NA indicates no data because all the mice died before 10 dpi. The scales in (**A**) were 50 µm, and the scales in (**B**) were 25 µm.

**Figure 5 ijms-25-00997-f005:**
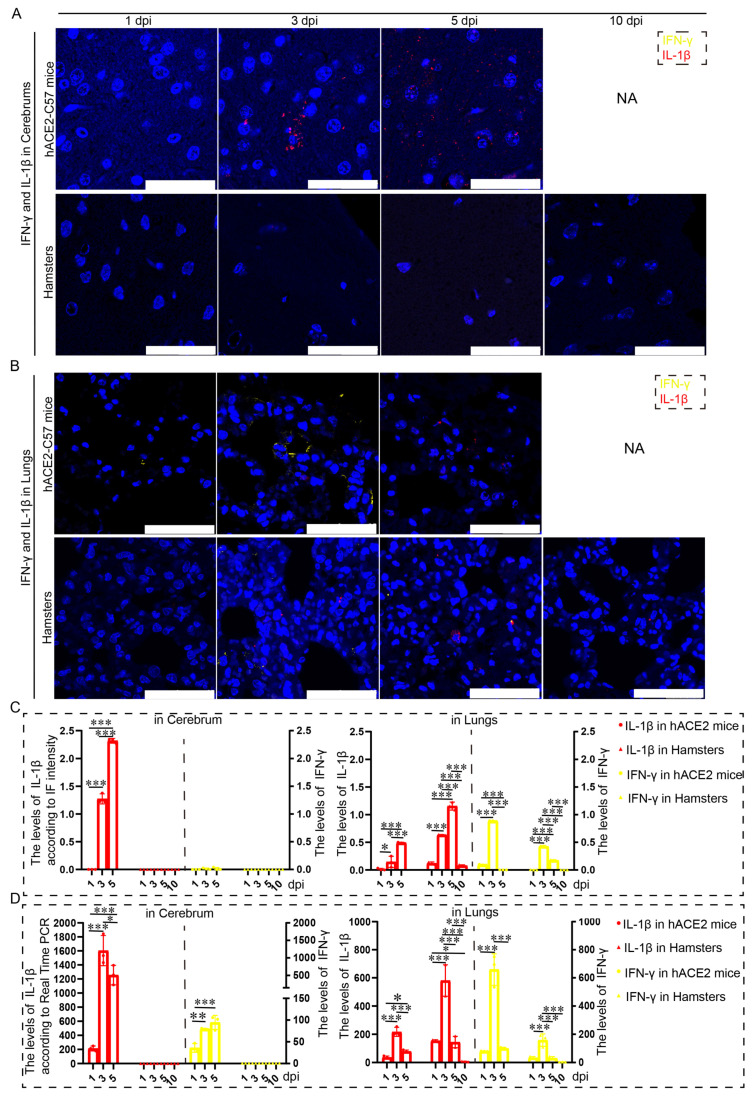
Both animals exhibited significant increases in IL-1β and IFN-γ expression in the lungs, but noticeable increases in IL-1β were observed in the cerebrum of hACE2-C57 mice. (**A**,**B**) The distribution and abundance of proinflammatory molecules, including IFN-γ and IL-1β, were detected via immunofluorescence in the cerebrum (**A**) and lungs (**B**) of hACE2-C57 mice and hamsters during SARS-CoV-2 infection. IFN-γ is shown in yellow, and IL-1β is shown in red. NA indicates no data because all the mice died before 10 dpi. The scales in the pictures were 100 µm. (**C**) The levels of IFN-γ and IL-1β in the cerebrum (**A**) and lungs (**B**) were calculated according to fluorescence intensity (*n* = 3). The values of IL-1β are shown on the left Y axis, and the values of IFN-γ are shown on the right Y axis. (**D**) The mRNA expression levels of IFN-γ and IL-1β in the cerebrum and lungs were detected via real-time PCR during SARS-CoV-2 infection (*n* = 3). The values of IL-1β are shown on the left Y axis, and the values of IFN-γ are shown on the right Y axis. * 0.01 < *p* ≤ 0.05, ** 0.001 < *p* ≤ 0.01, *** *p* ≤ 0.001. The error bars were calculated via two-way ANOVA using GraphPad Prism 8, and *p* values were calculated via one-way ANOVA using SPSS PASW statistical software version 18.0.

**Figure 6 ijms-25-00997-f006:**
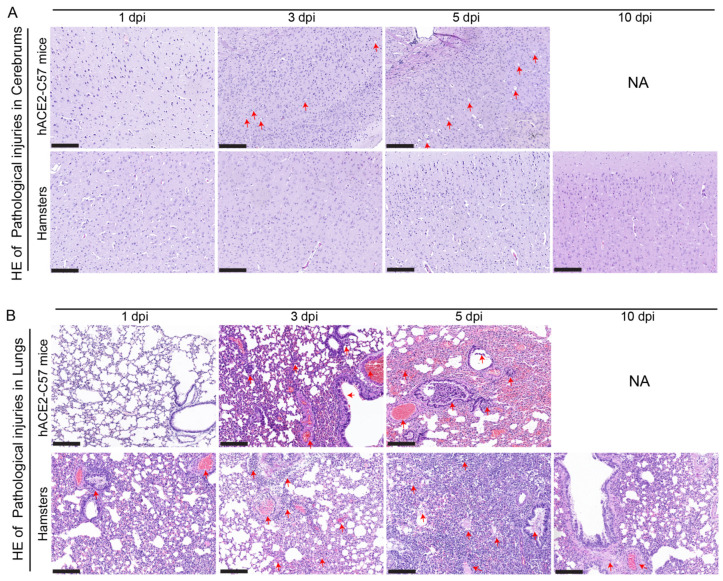
Pathological injuries were observed in the cerebrums of hACE2-C57 mice, while both animal models displayed severe interstitial pneumonia during SARS-CoV-2 infection. (**A**, **B**) Pathological injuries, including inflammatory cell infiltration, necrocytosis and blood extravasation, were detected and analyzed via HE staining in the cerebrum (**A**) and lungs (**B**) of hACE2-C57 mice and hamsters during SARS-CoV-2 infection. NA indicates no data because all the mice died before 10 dpi. The important and representative points with pathological damage are clearly indicated by red arrows. The scales in the pictures were 200 nm.

## Data Availability

All the data generated or analyzed during this study are included in this published article, and additional files are available from the corresponding author upon reasonable request.

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
