# Peer review of "The Abundant Distribution and Duplication of SARS-CoV-2 in the Cerebrum and Lungs Promote a High Mortality Rate in Transgenic hACE2-C57 Mice"

_ijms, 2024, doi:10.3390/ijms25020997_

Round 1

Reviewer 1 Report

Comments and Suggestions for Authors

In this manuscript, Li et al., investigated the SARS-0Cov-2 pathogenesis both in ACE2-C57 mice and hamster model. They showed that virus was distributed in the cerebral cortex neuron area and lung in mice. While virus was distributed in lung in hamster. They also found that SARS-CoV-2 induced significant increases in IL-1β and IFN-γ were observed in the lungs in both animals. Although this paper shows interesting difference between mouse and hamster, some control is missing and data is hard to convince. I have some comments on this manuscript.

1.     The authors should add the Figure numbers in the text (Results).

2.     Figure 1C and related methods: The authors should add more detail how they infected (i.n.?) and collected the sample. The virus shedding looks higher in early time point. Is it possible that they detected the input virus, not shedding?

3.     Figure 1C and D: In several organs, the viral loads are 4~6 (log10). Do the authors think this is positive or negative for SARS-CoV-2? In another word, which value is the limited detection? The authors should show the viral loads in uninfected animal.

4.     Figure 2A and material methods (2.4): Did the authors use same antibody to detect ACE2 both in mouse and hamster (this Ab recognize both mouse and hamster ACE2)? What is the 2nd antibody? The authors should clarify the information.

5.     Figure 2: The western blotting showed that ACE2 was expressed well in hamster lung and nasal mucosa. However, there were no ACE2 staining by IF in hamster. Can the authors explain this discrepancy? 

6.     Regarding the question above, ACE2 was expressed well in all organs except heart in mouse (by western blotting). However, there were no ACE2 staining by IF. The authors should describe the reason.

7.     Figure 3: Which color (blue or brown dot) is N staining? It’s much better to show as arrow.

8.     Figure 3: What the blue and brown color shows? The authors should clarify in the Figure legend.

9.     Figure 4: I didn’t see purple color in the Figure. It’s much better to show it as arrow. In addition, the word in the Figure is too small to read.

10.  Figure 5: IL-1b was upregulated in ACE2-mice, not in hamster. This should be important funding. The authors should discuss more what this means. They also need to cite some papers about IL-1b expression in SARS-CoV-2 infected model.

11.  Figure 5: The authors described that IL-1b and IFN were expressed both in mouse and hamster. However, it’ shard to see the signal in hamster. Is this real signal? Did the authors do some statics analysis? 

Comments on the Quality of English Language

Moderate editing of English language required.

Author Response

  1. The authors should add the Figure numbers in the text (Results).

Thank you very much for your advice, I am very sorry for the mission, and I have added the Figure numbers in the text in the revised manuscript.

  1. Figure 1C and related methods: The authors should add more detail how they infected (i.n.?) and collected the sample. The virus shedding looks higher in early time point. Is it possible that they detected the input virus, not shedding?

Thank you very much for your advice, and I have described the methods of infection mode and sampling collection in details in “2.1. Animals and Biosafety” of Materials and Methods in the revised manuscript. After anesthesia, the animals were infected by nasal drip; the drip for hACE2 mice was 20 µL, and that for hamsters was 80 µL. The nasal cavities were washed 3 times with an equal volume of PBS after 12 hours of dripping. And oropharyngeal shedding was assessed via swabs, while nasal shedding was assessed via 20 µL nasal washes through a micropipette, 100 mg homogenized tissue were used for the viral load detection by Real time PCR.

We found that all the virus shedding were highest at 2 dpi in the 4 infection groups, and the levels of virus shedding were also high at 1 dpi. In fact, both “Pathogenesis and transmission of SARS-CoV-2 in golden hamsters. Sia, S.F.; Yan, L.M.; and et al. Nature 2020, 583, 834-838, doi:10.1038/s41586-020-2342-5” and “Nasal Mucosa Exploited by SARS-CoV-2 for Replicating and Shedding during Reinfection. Li, H.; Zhao, X.; and et al. Viruses 2022, 14, doi:10.3390/v14081608” showed that the levels of viral shedding in nasal washes were high in the early point, and the levels of shedding in our manuscript were similar with them. And in our experimental methods, the nasal cavities were washed 3 times by equal volume of PBS after 12 hours of virus dripping. In our results, the levels of nasal shedding at 2 dpi increased by more than 10-fold than ones at 1 dpi, so high levels of viral duplication and shedding could be validated in early points. According to these data, we concluded that the major viral load in the nasal washed in the early points were from the viral shedding, although there may be a little input virus in the detections.

  1. Figure 1C and D: In several organs, the viral loads are 4~6 (log10). Do the authors think this is positive or negative for SARS-CoV-2? In another word, which value is the limited detection? The authors should show the viral loads in uninfected animal.

Thank you very much for your question, and according to the data in “Pathogenesis and transmission of SARS-CoV-2 in golden hamsters. Sia, S.F.; Yan, L.M.; and et al. Nature 2020, 583, 834-838, doi:10.1038/s41586-020-2342-5”, obvious viral RNA was detected at 7 dpi in hamsters while low levels of viral titers in the negative levels were detected. And the levels of viral RNA in tissues and viral shedding were similar in our last paper-“Nasal Mucosa Exploited by SARS-CoV-2 for Replicating and Shedding during Reinfection. Li, H.; Zhao, X.; and et al. Viruses 2022, 14, doi:10.3390/v14081608”. Here, we found that the viral loads are 4~6 (log10) in several organs at 10 dpi, and these were consistent with the data in our last paper, and there, we found that low levels of viral titers in the negative levels were detected although the mRNA of viral loads were about 4 (log10). Indeed, in uninfected animals, there was no viral load in all the tissues, and the Ct values were more than 35 in the real time PCR. As in our last paper, we concluded the clearance of viral mRNA were very slow in hamsters, and the viral mRNA was present and the viral loads are about 4 (log10) at 14 dpi and 21 dpi. In fact, the viral mRNA could be also detected with low levels at 10 dpi by RNASCOPE and immunofluorescence. In a word, viral mRNA could be detected with low levels about 4* log10 copies, but no viral titer appeared there.

  1. Figure 2A and material methods (2.4): Did the authors use same antibody to detect ACE2 both in mouse and hamster (this Ab recognize both mouse and hamster ACE2)? What is the 2ndantibody? The authors should clarify the information.

Thank you very much for your advice, and we used the same antibody (ab108209; Abcam) to detect ACE2 in both transgenic mice and hamster, and the antibody could not recognize the ACE2 in regular mice. The ACE2 in transgenic mice were humanized and they could be detected by ab108209. In addition, we found that the antibody could recognize the ACE2 in hamsters, maybe the related sequences were conserved between humans and hamsters. In western blot, goat anti-rabbit IgG (HRP) (ab6721) was used for visualization, and the details were described in Materials in the revised manuscript.

  1. Figure 2: The western blotting showed that ACE2 was expressed well in hamster lung and nasal mucosa. However, there were no ACE2 staining by IF in hamster. Can the authors explain this discrepancy? 

Thank you very much for your question, and in fact, obvious levels of ACE2 staining by IF in hamsters present in lung and nasal mucosa in Figure 2C and 2D, and indeed, the signals of ACE2 in hamsters were weaker than those in transgenic mice. ACE2 staining were shown in purple in Figure 2, and in the revised manuscript, we have added corresponding arrows in purple in Figure 2.

  1. Regarding the question above, ACE2 was expressed well in all organs except heart in mouse (by western blotting). However, there were no ACE2 staining by IF. The authors should describe the reason.

Thank you very much for your question, and in fact, obvious levels of ACE2 staining by IF in transgenic mice have been validated in all organs except heart, and ACE2 staining in cerebrum, lungs and nasal mucosa were shown in purple in Figure 2B, 2C and 2D, and we have added corresponding arrows in purple in Figure 2 in the revised manuscript. ACE2 staining in kidney, small intestine, liver and spleen were shown in purple in Figure s2.

  1. Figure 3: Which color (blue or brown dot) is N staining? It’s much better to show as arrow.

Thank you very much for your advice, and we are very sorry for the mission in the legend. N staining was shown in brown in the IHC staining, and we added the descriptions in figure legend. Please forgive that we did not add the arrows in the pictures due that the staining signals were too widespread.

  1. Figure 3: What the blue and brown color shows? The authors should clarify in the Figure legend.

Thank you very much for your advice, and we are very sorry for the mission. N staining was visualized in brown in the IHC staining because of the reaction between HRP and DAB, and the cell nucleus were visualized in blue because of the hematoxylin staining in cell nucleus. We added the descriptions in Figure 3 legend in the revised manuscript.

  1. Figure 4: I didn’t see purple color in the Figure. It’s much better to show it as arrow. In addition, the word in the Figure is too small to read.

Thank you very much for your advice, and we are very sorry for the dims in the pictures. The Probe-S-sense reflects viral duplication at the RNA level and is shown in purple, and the signals presented point by point and they were not pieces. In the revised manuscript, we have added corresponding arrows in purple (Probe-S-sense) or red (Probe-S) in the pictures, and the word in Figure 4 were larger. In addition, the pictures were constituted from 4 subpictures including Probe-S-sense signals in purple, Probe-S signals in red, Antibody-N signals in green and DAPI signals in blue, so the colors in the pictures were mixtures from them, and the colors would change when more than 2 sub colors presented in the same point.

  1. Figure 5: IL-1b was upregulated in ACE2-mice, not in hamster. This should be important funding. The authors should discuss more what this means. They also need to cite some papers about IL-1b expression in SARS-CoV-2 infected model.

Thank you very much for your advice, and we have added more evidences in Result, described the discussion in details and cited corresponding reference in the revised manuscript. In the revised manuscript, IL-1β and IFN-γ were detected by immunofluorescence and real-time PCR in the cerebrum and lungs of hACE2-C57 mice and hamsters, and we found that obvious IL-1β was present at 3 dpi early during infection and subsequently increased, and IL-1β reached its highest level in the cerebrum of hACE2-C57 mice until death; moreover, no obvious IL-1β was detected during end-stage infection in the lungs of hamsters.

IL-1β is one of the most powerful proinflammatory cytokines against infection and regulates the expression of several molecules involved in inflammation. In patients infected with SARS-CoV-2, increased levels of proinflammatory cytokines, including IL-1β, were found, and these increased cytokines and chemokines mediated infection immunopathogenesis and played important roles in the progression of COVID-19. An increase in IL-1β expression during SARS-CoV-2 infection was demonstrated in the serum of hACE2 aged mice and in the nasal mucosa of hamsters. Interestingly, increased IL-1β expression during SARS-CoV-2 infection was found in both the lungs and brain of newly weaned hamsters, and neuroinvasion of SARS-CoV-2 was demonstrated. We found clear expression of IL-1β early during infection at 3 dpi, but these levels increased later; IL-1β reached its highest level in the cerebrum of hACE2-C57 mice before death, and no obvious increase in IL-1β was detected in the hamsters. In addition, significant increases in IL-1β and IFN-γ were observed in the lungs in both animals. We concluded that IL-1β may play important roles in the inflammatory response in the cerebrum and lungs in SARS-CoV-2 infection, and high levels of viral distribution, IL-1β-associated inflammation and conspicuous pathological injuries in the cerebrum may promote susceptibility to SARS-CoV-2 infection and death in hACE2-C57 mice.

  1. Figure 5: The authors described that IL-1b and IFN were expressed both in mouse and hamster. However, it’ shard to see the signal in hamster. Is this real signal? Did the authors do some statics analysis?

Thank you very much for your question, and in the revised manuscript, we have calculated the levels of IFN-γ and IL-1β in the cerebrum and lungs according to fluorescence intensity from 3 biological parallels, and the mRNA expression levels of IFN-γ and IL-1β were detected by real-time PCR in the cerebrum and lungs during SARS-CoV-2 infection. According to these data, we demonstrated that obvious IL-1β presented in the cerebrum of hACE2-C57 mice, and no obvious IL-1β was detected in hamsters. And IL-1β and IFN-γ were expressed in the lungs of both mouse and hamster. More analysis and discussion were described in the revised manuscript.

12. Moderate editing of English language required.

Thank you very much for your advice, and our manuscript have been edited in www.aje.cn for 2 times, and the English language also have been edited by our specialist with extensive writing experience.

Reviewer 2 Report

Comments and Suggestions for Authors

In this manuscript Heng Li an co-authors describe the effects of SARS-CoV-2 infection in hamsters and transgenic hACE2 C57 mice, that are both susceptible to the infection but develop different pathologies: while hamsters recover and survive, ACE2 mice die within few days with both lung and brain damage. Molecular data on virus presence in different districts and the inflammation induced are provided.

My main concern about the manuscript is its lack of novelty: neural dissemination of SARS-CoV-2 in transgenic mice, its port of entry and pathogenesis have been reported multiple times; see for instance:  Carossino M et al. “Fatal Neurodissemination and SARS-CoV-2 Tropism in K18-hACE2 Mice Is Only Partially Dependent on hACE2 Expression”, 2022.

I am less aware of publications involving hamsters, but apparently, they are not scarce; one of the newest appears to be: Can Li et Al. “Intranasal infection by SARS-CoV-2 Omicron variants can induce inflammatory brain damage in newly weaned hamsters”, 2023.

In this work hamsters and mice were used in parallel, this strengthens the results, but that alone doesn’t seem enough to me. I suggest the authors to highlight the original points of their work in a way that the manuscript can be better judged.

Other points:

-Immunofluorescence data do not allow the reader to easily quantify the events described, e.g. cytokine production, a more quantitative analysis would have completed the work done.

-Fig. 6. More markers should be added to the pictures to identify the important structures/cells.

Author Response

  1. In this work hamsters and mice were used in parallel, this strengthens the results, but that alone doesn’t seem enough to me. I suggest the authors to highlight the original points of their work in a way that the manuscript can be better judged.

Thank you very much for your question, and we have described this question in details in Discussion in the revised manuscript.

According to previous reports, neuroinvasion by SARS-CoV-2 has been demonstrated in K18-hACE2 mice, HFH4-hACE2 mice, knock-in mice and newly weaned hamsters in multiple previous reports, and SARS-CoV-2 can enter the nervous system by crossing the neural–mucosal interface in the olfactory mucosa according to studies of autopsy material. Interestingly, pathological damage in the brain was validated in K18-hACE2 mice by TEM. Consistently, lethality was also found only in humanized ACE2 transgenic mice including K18-hACE2 mice and HFH4-hACE2 mice, we concluded lethality was consistent with the viral distribution in the brain. However, the respiratory tract was still the main site of viral distribution in these mice, and infections in the brain were observed only in the deceased mice, suggesting that neuroinvasion is not widespread in these transgenic mice.

Interestingly, we generated a different animal model, C57BL/6Smoc-Ace2em3(hACE2-flag-Wpre-pA)Smoc mice, in which abundant distribution and duplication of SARS-CoV-2 were detected in both the cerebrum and lungs via direct evidence from RNAScope and IHC; moreover, proinflammatory cytokines and pathological damage in the cerebrum were detected in situ. Consistent with previous data, infection in the cerebrum may be the major factor promoting lethality, given that all the transgenic hACE2-C57 mice died at high or low infection doses. Importantly, we found that high levels of IL-1β were present in the cerebrum after SARS-CoV-2 infection in transgenic mice, while the levels were low in hamsters. We concluded that IL-1β may play important roles in the inflammatory response and pathological damage in the cerebrum.

In fact, the major original points in our manuscript were the high lethality (100%) in the transgenic mice with high or low dose of SARS-CoV-2 infection and the pathological damage and inflammation in situ in the cerebrum, we demonstrated that the abundant distribution and duplication of SARS-CoV-2 in the cerebrum and lungs promote a high mortality rate in the transgenic hACE2-C57 mice, and maybe the infection in the cerebrum was as the watershed in the mortality.

We also think that the inflammatory brain damage in the newly weaned hamsters was very interesting, and in our manuscript, the major locations of SARS-CoV-2 infection and pathological damage were respiratory tracts in adult hamsters. Hamsters and mice were used in parallel, and these strengthened the results and conclusion in the correlation between lethality and infection in cerebrum.

Other points:

  1. Immunofluorescence data do not allow the reader to easily quantify the events described, e.g. cytokine production, a more quantitative analysis would have completed the work done.

Thank you very much for your question, and in the revised manuscript, we have calculated the levels of IFN-γ and IL-1β in the cerebrum and lungs according to fluorescence intensity from 3 biological parallels, and the mRNA expression levels of IFN-γ and IL-1β were detected by real-time PCR in the cerebrum and lungs during SARS-CoV-2 infection. According to these data, we demonstrated that obvious IL-1β presented in the cerebrum of hACE2-C57 mice, and no obvious IL-1β was detected in hamsters. And IL-1β and IFN-γ were expressed in the lungs of both mouse and hamster. More analysis and discussion were described in the revised manuscript.

  1. FIG 6. More markers should be added to the pictures to identify the important structures/cells.

Thank you very much for your advice, and we have added more arrows in Figure 6 in the revised manuscript, and the important and representative points with pathological damage were obviously recognized.

In addition, our manuscript has been edited in www.aje.cn for 2 times, and the English language also have been edited by our specialist with extensive writing experience.

Round 2

Reviewer 1 Report

Comments and Suggestions for Authors

The authors answered all my questions. I have no additional questions on this manuscript.

Comments on the Quality of English Language

Minor editing and final proof are required for final publication.

Author Response

Thank you very much for your time to review our manuscript, and one more inspection of our manuscript has been checked. We have added the methods and primers of Real-time PCR in “Materials and Methods”, and the related descriptions in details of statistical significance were added in “Materials and Methods” and Figure 5 in the revised manuscript.

Reviewer 2 Report

Comments and Suggestions for Authors
  1. Immunofluorescence data do not allow the reader to easily quantify the events described, e.g. cytokine production, a more quantitative analysis would have completed the work done.

Thank you very much for your question, and in the revised manuscript, we have calculated the levels of IFN-γ and IL-1β in the cerebrum and lungs according to fluorescence intensity from 3 biological parallels, and the mRNA expression levels of IFN-γ and IL-1β were detected by real-time PCR in the cerebrum and lungs during SARS-CoV-2 infection. According to these data, we demonstrated that obvious IL-1β presented in the cerebrum of hACE2-C57 mice, and no obvious IL-1β was detected in hamsters. And IL-1β and IFN-γ were expressed in the lungs of both mouse and hamster. More analysis and discussion were described in the revised manuscript.

thank you for your reply, but I could not find anything regarding PCR for cytokines in the materials and met. section.
Please, include also the number of animals tested for each experiment in the text or the captions and the statistical significance, if present.

Author Response

Thank you very much for your question, and we are very sorry for the mission of methods of the Real-time PCR. We have added the methods and primers of Real-time PCR in “Materials and Methods”. Both the numbers of animals tested in the calculation of IF intensity and mRNA expression levels were 3, and the statistical significance were calculated by SPSS, the related descriptions in details were added in “Materials and Methods” and Figure 5 in the revised manuscript.

Round 3

Reviewer 2 Report

Comments and Suggestions for Authors